# Human Biomonitoring Guidance Values (HBM-GVs) for Bisphenol S and Assessment of the Risk Due to the Exposure to Bisphenols A and S, in Europe

**DOI:** 10.3390/toxics10050228

**Published:** 2022-04-29

**Authors:** Matthieu Meslin, Claire Beausoleil, Florence Anna Zeman, Jean-Philippe Antignac, Marike Kolossa-Gehring, Christophe Rousselle, Petra Apel

**Affiliations:** 1French Agency for Food, Environmental and Occupational Health & Safety, Anses, 14 Rue Pierre et Marie Curie, 94701 Maisons-Alfort, France; matthieu.meslin@anses.fr (M.M.); claire.beausoleil@anses.fr (C.B.); 2French National Institute for Industrial Environment and Risks (INERIS), Parc ALATA BP2, 60550 Verneuil en Halatte, France; florence.zeman@ineris.fr; 3Oniris, National Research Institute for Agriculture, Food and the Environment (INRAE), LABERCA, 44300 Nantes, France; jean-philippe.antignac@oniris-nantes.fr; 4German Environment Agency (UBA), Corrensplatz 1, 14195 Berlin, Germany; marike.kolossa@uba.de (M.K.-G.); petra.apel@uba.de (P.A.)

**Keywords:** human biomonitoring (HBM), HBM4EU, internal exposure, biomarkers, endocrine disruptors, bisphenol A (BPA), bisphenol S (BPS), bisphenols, human biomonitoring guidance value (HBM-GV), physiologically based pharmacokinetic modelling (PBPK), risk assessment

## Abstract

Within the European Joint Programme HBM4EU, Human Biomonitoring Guidance Values (HBM-GVs) were derived for several prioritised substances. In this paper, the derivation of HBM-GVs for the general population (HBM-GV_GenPop_) and workers (HBM-GV_worker_) referring to bisphenol S (BPS) is presented. For the general population, this resulted in an estimation of the total urinary concentration of BPS of 1.0 µg/L assuming a 24 h continuous exposure to BPS. For workers, the modelling was refined in order to reflect continuous exposure during the working day, leading to a total urinary concentration of BPS of 3.0 µg/L. The usefulness for risk assessment of the HBM-GVs derived for BPS and bisphenol A (BPA) is illustrated. Risk Characterisation Ratios (RCRs) were calculated leading to a clear difference between risk assessments performed for both bisphenols, with a very low RCR regarding exposure to BPA, contrary to that obtained for BPS. This may be due to the endocrine mediated endpoints selected to derive the HBM-GVs for BPS, whereas the values calculated for BPA are based on the temporary Tolerable Daily Intake (t-TDI) from EFSA set in 2015. A comparison with the revised TDI recently opened for comments by EFSA is also discussed. Regarding the occupational field, results indicate that the risk from occupational exposure to both bisphenols cannot be disregarded.

## 1. Introduction

Bisphenols are a group of chemical compounds with two hydroxyphenyl functionalities. BPA, the most commonly used compound of the bisphenol family, is a monomer mainly used in the production of polycarbonate plastics and epoxy resins [1]. Polycarbonates can be used in the manufacture of many consumer items, such as food containers, baby bottles and childcare articles, water pipes, etc. Epoxy resins are used to make protective coatings for cans and beverages, and to make coatings used on the metal lids of jars [2]. Any residual BPA present in the final material or article made from polycarbonate plastics or epoxy resins has the potential to migrate into the food or water with which it comes into contact [3]. BPA is also found in air and dust particles and is used in some paper products (thermal paper, e.g., sales receipts) as well as in many other products, such as medical devices, surface coatings, printing inks, dental sealants, toys, cosmetics and flame retardants [2,4,5,6,7]. Human exposure to BPA is thus widespread, and ingestion of contaminated food is a major contributor to overall internal exposure to BPA for all age groups [8].

The toxicity of BPA has been extensively characterised in several risk assessments carried out by different bodies such as EFSA (the European Food Safety Authority), ANSES (the French Agency for Food, Environmental and Occupational Health & Safety, Maisons-Alfort, France) and even the Food and Agriculture Organization and the US Food and Drug Administration [2,9,10,11]. The German HBM Commission also reviewed BPA in 2012 and established HBM-I values for children and adults to assess HBM results [12,13]. These values were updated in 2015 on the basis of the new EFSA temporary Tolerable Daily Intake (t-TDI) available at that time. The latest comprehensive reassessment of BPA exposure and toxicity (by EFSA in January 2015 leading to its t-TDI of 4 µg/kg bw/d [2]) is based on the increase in relative kidney weight in the F0 generation of a two-generation mouse study by Tyl et al. (2008) seen as a critical effect [14]. According to EFSA, the risk to consumers was controlled as the highest estimates for dietary exposure and for exposure from several sources, both oral and dermal, are 3–5 times lower than the new TDI [2]. In 2015, the Risk Assessment Committee (RAC) of the European Chemicals Agency (ECHA) also issued an opinion following a request for the restriction of BPA in thermal paper. The restriction dossier was submitted by the French authorities in May 2014, following the identification of health risks for unborn children linked to the exposure (occupational or not) of their mothers to BPA contained in these thermal papers [15]. The RAC followed the same approach as that used by EFSA to calculate an oral Derived No Effect Level (DNEL) for the general population. Since the proposed restriction targeted dermal exposure when handling thermal paper, DNELs for dermal exposure (assuming 50% bioavailability) were also calculated for workers and the general population [15]. Based on these dermal DNELs and reasonable “worst case” scenario modelling, the RAC concluded that the risk of exposure to BPA from handling thermal paper was controlled for consumers but insufficiently controlled for workers. On 12 December 2016, the European Commission amended Annex XVII of REACH to include a restriction which set a threshold limit of 0.02% (by weight) for BPA in thermal paper. This restriction came into force on 2 January 2020 [16]. 

Nevertheless, endocrine disrupting properties of BPA at low doses are strongly suspected, and concern especially exists, for particularly sensitive population groups such as pregnant women and young children [17]. In January 2017, BPA was included on the candidate list of substances of very high concern (SVHC) due to its reproductive toxicity. In June of the same year, the ECHA Member State Committee supported the French proposal to include its endocrine disrupting properties, which cause probable serious effects on human health and give rise to a level of concern equivalent to carcinogenic, mutagenic, toxic to reproduction (CMRs) category 1A or 1B. In 2018, an update added an additional reason for inclusion in the candidate list of SVHC due to its endocrine disrupting properties leading to adverse environmental effects, as proposed by Germany [18].

The EFSA Panel on Food Contact Materials, Enzymes and Processing Aids (CEP Panel) released for a draft opinion for public consultation in December 2021 on the re-evaluation of risks to public health due to the presence of BPA in foodstuffs [19]. In this draft opinion, the immune system has now been identified as the most sensitive health endpoint to BPA exposure. Specifically, an increase of Th17 cells, key players in cellular immune mechanisms and involved in the development of allergic lung inflammation, was identified as the critical effect. This work led the EFSA CEP Panel to reduce the previous t-TDI by an order of magnitude of 10^5^, which leads to a value of 0.04 ng/kg bw/day. However, as this opinion remains at this stage a draft, the risk assessment performed in this paper will still consider the t-TDI set in 2015.

Bisphenol S (BPS) is used in a variety of industrial applications as a substitute for BPA, for example, as a wash fixative in cleaning products, or as a starting monomer for the synthesis of polyether sulfone specifically used in the manufacture of baby bottles and children’s tableware. Available data on the presence of BPS in the environment are more limited than for BPA. In Europe, BPS was detected in the wastewater of industrial facilities [20] and canned food [21]. BPS is also used as a developer in thermal paper, including in products marketed as “BPA-free paper” [22]. A survey of manufacturers selling thermal paper in the EU conducted by ECHA indicates that the use of BPS as an alternative to BPA in the manufacture of thermal paper had not increased markedly for the period 2014–2016 [23]. However, it is important to continue to monitor which alternatives thermal paper manufacturers will prefer, following the restriction on the use of BPA in these papers. Indeed, a study in China examined the presence of thirteen bisphenol-related compounds in paper products (120 thermal papers and 81 non-thermal papers) collected in Beijing [24]. The results indicated that the replacement of BPA, by alternatives such as BPS, has progressed significantly in several types of thermal paper such as stickers, train tickets, aeroplane boarding passes, lottery tickets, etc. Due to the probably globalised production of these products, such an increase in Europe cannot be excluded. 

According to the Classification, Labelling and Packaging (CLP) regulation, BPS is considered a presumed reproductive toxicant for fertility and development (Reproductive Toxicant 1B). No risk assessment has yet been published for BPS, but several initiatives are underway. The toxicological profile of BPS has however been studied by Beausoleil et al. (2022) [25]. There is a fairly large body of data on the toxicity of BPS, including repeated dose toxicity, reproductive toxicity and also specific effects on the endocrine system. 

Human biomonitoring (HBM) is a tool of growing importance for estimating internal aggregated exposure to chemicals and can therefore be used to improve human health risk assessment. The European Joint Programme on Human Biomonitoring (HBM4EU) is a joint effort of 30 countries and the European Environment Agency (co-funded by the European Commission within the framework of Horizon 2020) with the aim of advancing and harmonising human biomonitoring in Europe [26]. One of the limitations for the use of HBM data in risk assessments (RA) is the lack of HBM guidance values (HBM-GV) [27] which represent the concentration of a biomarker of exposure in a human biological matrix at and below which no adverse effect for human health is expected according to current toxicological knowledge. Those values are useful to risk assessors and risk managers for comparing to the HBM data measured in biomonitoring studies. This paper aims to characterise the risks associated with BPS and BPA according to available HBM data in Europe. For this purpose, the derivation of HBM-GVs for exposure to BPS for both the general population (HBM-GV_GenPop_) and for workers (HBM-GV_Worker_) has also been performed, as has been done previously for BPA [28]. 

## 2. Materials and Methods

### 2.1. Characterisation of Risk

To estimate the risk due to BPA or BPS exposure, Risk Characterisation Ratios (RCRs) are calculated by comparing the 95th percentile (P95) of the levels of biomarker measured to the *HBM-GV* derived, as described in Equation (1).
(1)$$RCR=\frac{P95\;\left(\mathrm{biomarker}\right)}{HBM-GV}$$

If the RCRs are lower than 1, the risk due to BPA or BPS exposures can be ruled out for the sampled population according to the *HBM-GV* derived. Conversely, if RCRs are higher than 1, the risk cannot be ruled out for the health of the sampled population and investigations and/or management measures are required.

### 2.2. Derivation of HBM-GVs: General Methodology

The HBM-GVs are derived according to a systematic methodology agreed within the HBM4EU project and published in 2020 by Apel et al. [29]. 

In brief, this methodology relies on a tiered approach depending on the knowledge and data available (Figure 1). Option 1 is followed when relevant human data proving a relationship between internal concentrations of an adequate biomarker and the occurrence of adverse health effects are available. If those data are insufficient or unavailable, the second option can be considered by translating an existing external toxicity reference value (TRV), such as a TDI, or an Occupational Exposure Limit set by EU or relevant non-EU bodies, into an internal value. As a final option, HBM-GVs can be derived on the basis of critical effects observed in animal toxicological studies. This methodology has been applied for deriving HBM-GVs for BPA and BPS and the best approach was selected. 

### 2.3. Characterisation of Internal Exposure to BPA and BPS

The Information Platform for Chemical Monitoring (IPCheM) was consulted to obtain HBM data from European cohorts. IPCheM is a platform developed under the initiative of the European Commission for accessing chemical data collected and managed in Europe [30]. P95 for the selected biomarkers (total BPA and total BPS) in urine from aggregated HBM data included in the IPCheM platform was considered for the risk assessment of BPA and BPS. To respect data ownership, only results published in the literature were used in this paper. To document BPA exposure for the occupational population, data from occupational studies included in the review from Bousoumah et al. (2021) [31] were also used. 

## 3. Results

### 3.1. Assessment of Risk Due to BPA Exposure

#### 3.1.1. Derivation of HBM-GVs for BPA

##### Derivation of HBM-GV_GenPop_

The HBM-GV_GenPop_ for adults and children was developed for total urinary BPA and published by Ougier et al. (2021) [28]. Human data proving a relationship between internal concentrations of a specific and sensitive biomarker of BPA and the occurrence of relevant adverse health effects were not sufficient to derive a HBM-GV. Thus, the second option for HBM-GV derivation, relying on the conversion of an external guidance value into a corresponding internal level, was adopted. For this purpose, total BPA concentrations in urine were estimated considering a steady-state exposure at the EFSA t-TDI of 4 µg/kg bw/d, using the PBPK model developed by Karrer et al. (2018) [32], assuming a 24 h averaged BPA exposure that is constant and occurs 100% via the oral route. The resulting values derived by Ougier et al. (2021) are 230 µg/L for adults and 135 µg/L for children [28] (Table 1). 

##### Derivation of HBM-GV_worker_

For the occupationally exposed population, the absorption of BPA results mainly from dermal exposure, in particular for cashiers. 

Therefore, the total BPA level in urine was also estimated by Ougier et al. (2021) for dermal absorption that would generate the same level of free BPA plasma concentration as an average 24 h oral absorption at the ECHA DNEL of 8 µg/kg bw/d for workers. A concentration of approximately 12 µg/L of total BPA in urine was calculated by Ougier et al. (2021) as a biological threshold value not to be exceeded in workers, assuming 100% dermal exposure in the workplace [28]. However, since this value for workers is much lower than the recommended HBM-GV_GenPop_ (230 µg/L) and lower than the total BPA concentrations from some of the environmental (non-occupational) background exposures identified, no HBM-GV_worker_ could be recommended. Thus, the theoretical threshold value of 12 µg/L corresponding to the worst-case scenario resulting from 100% dermal exposure was used only as an indication for comparison with available biomonitoring data.

#### 3.1.2. Assessment of Exposure to BPA

For the assessment of risks due to BPA exposure for the general population, data of 16 HBM studies from 16 different European countries, representative of different parts of Europe, were compiled (Table 2). Data from the HELIX cohorts were creatinine adjusted. To be able to compare the creatinine adjusted levels to the reported HBM-GVs, data were converted in unadjusted levels (µg/L) using the mean value for creatinine obtained from studies by MacPherson et al. (2018) and Lau et al. (2018) for adults and children, respectively [33,34]. Moreover, reported levels of total BPA in urine corrected by urine specific gravity could not be converted, and were used as they were for the calculation of the RCR. 

Regarding occupational exposure to BPA, HBM data are more limited and those identified in a review published by Bousoumah et al. (2021) from five European occupational studies are discussed for risk assessment purposes [31]. A Finnish study by Heinälä et al. published in 2017 and involving 49 workers was conducted in five different companies using BPA in the production of paints, composites, tractors and thermal papers [35]. Simonelli et al. (2017) collected urine samples from workers suffering endometriosis and a healthy control group, as well as information about their lifestyle, environment and work activities [36]. Lyapina et al. (2016) also conducted a study in Bulgaria in 55 students of dental medicine exposed to dental composite resins containing BPA as impurities due to their production process, and in 29 patients treated with those composite resins but not exposed occupationally as controls [37]. A Spanish study by Gonzales et al. (2019) has determined the levels of eight Bisphenols including BPA, BPS and BPF in biological samples from a controlled cohort of 29 workers in a hazardous waste incinerator located in Constantí in the Catalonia region [38]. Lastly, a French study performed by Ndaw et al. (2016) evaluated the exposure to BPA in cashiers and in non-occupationally exposed workers from several workplaces [39].

#### 3.1.3. Risk Characterisation Due to BPA Exposure 

##### For the General Population

RCRs were calculated by comparing total BPA levels in urine from the HBM studies to the corresponding HBM-GVs derived for BPA (Table 2). The t-TDI based HBM-GV_GenPop_ for total BPA in urine for adults was used for calculating RCRs for teenagers. For studies reporting results for children and teenagers in combination, the HBM-GV for children was used with a conservative approach. In all cases the calculated RCRs are significantly lower than 1, both for adults and for children, and for each set of data available, meaning that exposure levels reported are far lower than the HBM-GV_GenPop_. These results indicate that exposure to BPA for the populations sampled, according to current knowledge in a single substance–risk assessment approach, does not constitute a risk when taking into account the HBM-GV_GenPop_ derived from the t-TDI established by EFSA in 2015.

**Table 2 toxics-10-00228-t002:** Reported concentrations of total urinary BPA in the general population (P95) and RCR calculated using the HBM-GV_GenPop_ recommended by HBM4EU.

Cohorts	References	Country	Populations	P95 (µg/L)	HBM-GV_GenPop_ for Total Urinary BPA (µg/L)	RCR
IBS	Berman et al., 2014 [40]	Israel	Adults (40–59 years old)	20.48	230	0.09
GerES IV *	Becker et al., 2009 [41]	Germany	Children (3–14 years old)	14	135	0.10
GerES V	Tschersich et al., 2021 [42]	Germany	Children (3–5 years old)	7.79	135	0.06
			Children (6–10 years old)	5.13	135	0.04
			Children (11–13 years old)	9.95	135	0.07
			Adolescents (14–17 years old)	7.65	230	0.03
3xG	Study website [43]	Belgium	Adults (20–39 years old)	4.61	230	0.02
FLEHS II	Geens et al., 2014 [44]	Belgium	Adolescents (14–15 years old)	9.6	230	0.04
DEMOCOPHES	Covaci et al., 2015 [45]	Belgium, Denmark, Luxembourg, Slovenia, Spain, Sweden	Mothers (<45 years old)	11.1	230	0.05
			Children (5–12 years old)	13.1	135	0.10
		Belgium	Mothers (<45 years old)	11.6	230	0.05
			Children (5–12 years old)	13.4	135	0.10
		Denmark	Mothers (<45 years old)	11.5	230	0.05
			Children (5–12 years old)	7.9	135	0.06
		Luxembourg	Mothers (<45 years old)	7.4	230	0.03
			Children (5–12 years old)	8.3	135	0.06
		Slovenia	Mothers (<45 years old)	13.4	230	0.06
			Children (5–12 years old)	18.9	135	0.14
		Spain	Mothers (<45 years old)	12.2	230	0.05
			Children (5–12 years old)	9.8	135	0.07
		Sweden	Mothers (<45 years old)	5	230	0.02
			Children (5–12 years old)	6.2	135	0.05
PBAT	Hartmann et al., 2016 [46]	Austria	Children M (6–10 years old)	7.3	135	0.05
			Children F (6–10 years old)	5.8	135	0.04
			Teenagers M (11–15 years old)	2.7	230	0.01
			Teenagers F (11–15 years old)	5.2	230	0.02
			Adults M (18–64 years old)	3.6	230	0.02
			Adults F (18–64 years old)	1.3	230	0.01
			Senior M (65–79 years old)	1.3	230	0.01
			Senior M (65–79 years old)	4.6	230	0.02
HELIX **	Haug et al., 2018 [47]	France, Greece, Lithuania, Norway, Spain, United Kingdom.	Pregnant women (>18 years old)	22	230	0.10
			Children (6–12 years old)	15.7	135	0.12
Elfe	Dereumeaux et al., 2016 [48]	France	Pregnant women (>18 years old)	5.3	230	0.02
Esteban	Balicco et al., 2019 [49]	France	Children (6–10 years old)	7.3	135	0.05
			Adolescents (11–14 years old)	13.7	230	0.06
			Adolescents (15–17 years old)	6	230	0.03
			Adults M (18–74 years old)	10	230	0.04
			Adults F (18–74 years old)	6.9	230	0.03
Danish-HBM	Frederiksen et al., 2014 [50]	Denmark	Pregnant women (>18 years old)	7.52	230	0.03
RefLim 2011	Porras et al., 2014 [51]	Finland	Adults (22–67 years old)	7.9	230	0.03
RHEA	Myridakis et al., 2015 [52]	Greece	Pregnant women (>16 years old)	4.7	230	0.02
			Children (4 years old)	16.6	135	0.12
INMA	Casas et al., 2013 [53]	Spain	Pregnant women (≥16 years) first trimester	11.9	230	0.05
			Children (4 years old)	12.3	135	0.09
TH Pregnant women	Machtinger et al., 2018 [54]	Israel	Pregnant women	14.2	230	0.06
SLO-CRP	Tkalec et al., 2021 [55]	Slovenia	Children (6–9 years old)	9.5	135	0.07
			Adolescents (11–15 years old)	7.3	230	0.03

* The HBM-GV value for children was used to calculate the RCR using a protective approach. ** Conversion carried out with concentrations of 0.75 g creatinine/L for pregnant women (MacPherson et al., 2018) and 0.68 g creatinine/L for children (Lau et al., 2018).

##### For Workers

As explained above, no HBM-GV could be derived for workers exposed to BPA. However, the concentration of 12 µg/L of total BPA in urine calculated by Ougier et al. (2021) was used to compare the values measured in occupational studies [28]. The study by Heinälä et al. (2017) suggests an occupational exposure of concern, with median urinary concentrations (P50) of total BPA at the end of the shift much higher (130–150 µg/L) in an environment where the air levels of BPA are low [35]. Elevated levels of total urinary BPA were still observed on Monday morning after the weekend break. Taking into account this delay in the excretion of BPA, these results suggest significant skin exposure, for which absorption is generally slower in comparison with other routes of exposure. In addition, the highest reported concentrations of total BPA were in the order of 1000–1500 µg/L, which is well above the value of 12 µg/L, but also above the recommended HBM-GV for the general population (230 µg/L). 

Simonelli et al. (2017) reported urinary concentrations of total urinary BPA in workers lower than in the study of Heinälä et al. (2017), with concentrations ranging from 1.17 µg/L to 12.68 µg/L, and an average value of 5.31 µg/L [36]. In the control group, the concentrations measured are between 1.28 and 2.35 µg/L, with an average value of 1.64 µg/L. These levels are 10 to 100 times lower than those reported by Heinälä et al. (2017) and are also below, or at the limit of, the value of 12 µg/L while also well below the HBM-GV_GenPop_ value (230 µg/L). In addition, the results from Simonelli et al. (2017) show that certain professional activities generally considered to be exposed to BPA, such as those carried out by housekeepers, are in reality slightly exposed or not exposed to BPA. These observations could be due to the wearing of personal protective equipment. On the contrary, other categories generally considered to have a rather small exposure to BPA through their professional activities, such as students or salespeople, would in reality be significantly exposed.

Lyapina et al. (2016) and Gonzales et al. (2019) also reported results below the theoretical limit value of 12 µg/L [37,38]. The urinary concentration of total BPA was on average 6.16 µg/L, with a standard deviation of 14.05 µg/L, in student dentists, and only 0.86 µg/L for employees working in a hazardous waste incinerator with a maximum value of 2.82 µg/L. 

The study by Ndaw et al. published in 2016 reports median urinary concentrations of total BPA of 3.54 µg/L in controls and 8.82 µg/L in cashiers [39]. These levels are also below both the HBM-GV_GenPop_ and the value of 12 µg/L. However, the P95 values exceed this limit value for both exposed cashiers and for controls with 44 µg/L and 14.2 µg/L, respectively. However, the authors highlighted that no significant increase in the urinary concentrations of free BPA was observed.

### 3.2. Assessment of Risk Due to BPS Exposure

#### 3.2.1. Derivation of HBM-GVs for BPS

##### Selection of the Methodological Approach for the Derivation of HBM-GVs for BPS

The human data analysed by Beausoleil et al. (2022) were considered insufficient for establishing a relationship between internal concentrations and relevant health effects [25]. In addition, neither toxicity reference values nor occupational exposure limits have been proposed by the EU or by any relevant non-EU organisations so far. Regarding workers, the scarcity of data on occupational exposure does not allow for assessment of the relationship between atmospheric concentrations of BPS and total urinary concentrations of BPS. Thus, the derivation method for HBM-GVs for BPS for both the general population and the workers was based on a point of departure (POD) identified from animal experimental studies, as tentatively proposed by Beausoleil et al., 2022 [25]. The HBM-GVs were then derived by translating the POD into the corresponding urinary levels in humans for the selected biomarker of exposure by using a PBPK model. Once the human equivalent dose was derived, different assessment factors were applied to estimate the HBM-GVs. 

Karrer et al. (2018) extended to BPS the PBPK model developed by Yang et al. in 2015, used previously by EFSA in 2015 for calculating the Human Equivalent Dose Factor (HEDF) (ratio of AUC Animal/AUC Human) and deriving the t-TDI for BPA [2,32,56]. To validate the model, a comparison of predicted concentrations with measured concentrations of BPS and total BPS (BPS-G + free BPS) in two volunteer studies [57,58,59] has been performed. The blood flow was replaced by a plasmatic flow (BPS being mostly distributed in the plasma) and a value for the haematocrit of 0.45 was used. Outputs of excreted urinary BPS quantities were also replaced with urinary BPS concentrations by adding an estimated value for the daily 24 h urinary excretion. The same daily urinary excretion rate of 0.05 L/h previously used for BPA was considered. After oral administration, an absorption fraction of 100% was considered. 

BPS is well absorbed by the oral route. However, dermal absorption of BPS is very limited. Based on an in vitro study by Champmartin et al. (2020) [60], it has been assumed that the dermal route will contribute very little to the overall exposure to BPS. Lastly, the relative contribution of the inhalation route may also be expected to be low, based on BPS physical-chemical properties (low vapor pressure) and by analogy with previous BPA risk assessments. It is thus anticipated that the 100% oral exposure scenario is appropriate for the general population, as well as for workers. Thus, the derivation method for HBM-GVs for BPS for both the general population and workers is based on the same constant and continuous exposure scenario to the LOAEL chosen as a POD via ingestion (100% oral exposure scenario).

##### Selection of the Biomarker of Exposure for BPS

Bisphenols are largely detoxified by phase II conjugating enzymes, including UDP-glucuronyl transferases (UGT) and sulfotransferases (SULT) (Figure 2) [61,62]. Similarly to what has been observed for BPA, BPS clearance is mainly driven by glucuronidation of BPS into BPS-G, principally excreted in urine [28]. Although the unconjugated form is known to be the active one able to induce an effect, very few data are available regarding the excretion levels of non-conjugated BPS in urine; however, these levels confirmed a very low proportion of free urinary BPS among all the excreted forms [49]. Conversely, the measurement of total BPS in urine allows for better compatibility with current methodological detection limits and is a protective approach for exposure assessment considering possible endogenous deconjugation. Therefore, in a large majority of available biomonitoring studies, the urinary exposure level is estimated after enzymatic hydrolysis of the conjugates so that the total of free and conjugated forms is finally determined. Total BPS in urine was then selected as the appropriate biomarker of exposure to BPS to be considered for the derivation of HBM-GV. It should be noticed that most of the existing studies have determined exposure levels in spot urine samples.

##### Selection of Key Studies and Choice of POD

The first two options that, according to methodology developed within the HBM4EU programme and published by Apel et al. (2020), should be used to derive the HBM-GV when the data available allows it, could not be followed for BPS. Therefore, the third option is to derive an HBM-GV based on critical effects observed in animal studies. The choice of POD is based on the recommendation formulated by Beausoleil et al. (2022) who reviewed regulatory toxicology studies conducted in accordance with OECD Test Guidelines, as well as numerous academic studies investigating more specific parameters dedicated to endocrine disrupting effects [25]. Adverse effects due to BPS exposure were reported in academic studies assessing male and female reproduction, mammary glands, neurobehavior and metabolism/obesity. LOAELs regarding developmental exposure to BPS were far lower in academic studies than in regulatory ones and were expressed, respectively, in µg/kg bw/day vs. mg/kg bw/day. Beausoleil et al. (2022) proposed to derive a HBM-GV from values determined in academic peer-reviewed experimental studies [25]. 

Converging studies by Kolla et al. published in 2018 and 2019, as well as by Catanese and Vandenberg published in 2017, were conducted by exposure via the oral route with at least two dose levels, and identified a common LOAEL of 2 µg/kg bw/day for mammary gland and neurobehavioral toxicity, respectively (Table 3) [63,64,65]. This LOAEL has been selected as a POD for the derivation of the HBM-GVs.

##### Prediction of the Total Urinary BPS Concentration with a 100% Oral Exposure Scenario

The derivation method for HBM-GVs for BPS for both the general population and workers is based on the same LOAEL identified from animal experimental studies. In addition, dermal absorption of BPS is very limited, whereas BPS is well absorbed by the oral route. It is therefore anticipated that the dermal route will contribute very little to the overall exposure to BPS. The relative contribution of the inhalation route may similarly be expected to be low, based on its physical-chemical properties (low vapor pressure) and by analogy with previous BPA risk assessments. Therefore, a 100% oral exposure scenario was considered for the general population, as well as for workers. 

The HBM-GVs were derived by translating the LOAEL of 2 µg/kg bw/day, chosen as a POD for the corresponding urinary levels in humans of total BPS, by using the modified PBPK model published by Karrer et al. in 2018 [32]. Moreover, an assessment factor of 3 is applied for extrapolating a LOAEL to a NOAEL. The interspecies differences in toxicokinetics between animal species and humans are determined by PBPK modelling, and an assessment factor of 2.5 is applied for remaining interspecies differences (mostly for toxicodynamic differences). A factor of 10, accounting for intra-species differences, is further applied to the human adjusted NOAEL. Generally, this factor is set to 5 when workers are the targeted population for which the HBM-GV is derived, in line with ECHA’s R8 guidance for deriving DNELs for workers [66]. However, as the most sensitive endpoint(s) to be protected from are the effects on the unborn child, no differences can be assumed between the foetuses of the general population and those of workers.

Therefore, for an adult of 70 kg, applying assessment factors of 75 (3 × 2.5 × 10) to the POD and assuming a constant and continuous oral exposure throughout the day of BPS leads to a constant exposure to 0.0266 µg/kg bw/day. The estimated maximal concentration of plasmatic free and glucuronidated BPS is 9.7 × 10^−3^ nmol/L (2.42 ng/L) after 24 h constant oral exposure to 0.0266 µg/kg bw/day for an adult of 70 kg (Figure 2).

The estimated concentration of total BPS (sum of estimated urinary glucuronidated and free BPS) in urine is 4.13 nmol/L (1030 ng/L) after 24 h constant oral exposure to 0.0266 µg/kg bw/day for an adult of 70 kg (Figure 3). The HBM-GV_GenPop_ is then rounded to 1.0 µg/L.

Although a weight of 70 kg may not be representative of women, especially non-pregnant women, this weight gives more conservative values for lighter women and seems appropriate for calculating HBM-GVs for a population including men and women. Additionally, as the most sensitive endpoint(s) to be protected from are the effects on the unborn child, the calculated HBM-GVs for men are probably highly conservative. 

For workers, the same approach of using the POD and modified PBPK model by Karrer et al. (2018) was used to derive the HBM-GV_worker_. Applying an assessment factor of 75 (3 × 2.5 × 10) and assuming a constant and continuous oral exposure of BPS throughout the day would lead to a constant exposure of 0.0266 µg/kg bw/day. However, as occupational exposure is not continuous throughout the day or week, the modelling was refined in order to reflect a continuous exposure of 8 h per day followed by a non-exposure period of 16 h. This 8 h exposure and 16 h non-exposure period was then repeated four times in order to mimic a working week. Thus, this 5 day occupational scenario includes an 8 h exposure period with 0.0798 µg/kg bw and a non-exposure period of 16 h which then corresponds to an exposure for the entire day (0–24 h) of 0.0266 µg/kg bw/day (Figure 4 and Figure 5). The estimated concentration of total BPS (sum of estimated urinary BPS-G and free BPS) in urine is 12.1 nmol/L (3028 ng/L). The HBM-GV_worker_ is then rounded to 3.0 µg/L.

#### 3.2.2. Assessment of Exposure to BPS

The data from studies reporting total BPS levels in urine were used to calculate the RCRs with the corresponding HBM-GVs detailed above. For the risk assessment for the general population exposed to BPS, data from seven studies from six different European countries were compiled. RCRs were calculated for each available dataset, with the exception of studies for which P95 values were not reported. Some studies also reported results corrected by urine specific gravity. As with BPA, the results of those studies were used as is and even though they are close to the HBM-GVs this contributes to the uncertainty of the meaning of the corresponding RCRs calculated. In addition, unlike BPA, a specific HBM-GV_GenPop_ value for children was not developed for BPS and a general value calculated based on adult weight was used. Calculating such a value based on a lower weight that is more representative of a child population would have led to more conservative RCRs and the characterisation of a potentially increased risk for these populations.

Data on occupational exposure to BPS in Europe are more limited. In fact, given our literature search, only one of the retrieved studies describes the exposure assessment of European (French) workers to BPS and is related to thermal receipts. 

#### 3.2.3. Risk Characterisation Due to Exposure to BPS

##### For the General Population

Taking into account the HBM-GV_GenPop_ developed, and the internal exposure measured in the Esteban study, the RCRs obtained with the P95 are greater than 1 for all age groups (Table 4) [42]. Therefore, the health risk associated with exposure to BPS cannot be excluded. Looking at other percentiles than P95, we find that this risk exists for at least 10% of the sampled population. Regarding the Israeli study by Machtinger et al. (2018) [54], the RCR is less than 1, which means that the health risk due to BPS exposure of the studied population can be ruled out with respect to the HBM-GV_GenPop_. According to the result of the Slovenian study by Tkalec et al. (2021) [55], BPS exposure is not a risk for children in contrast to the adolescents sampled in this study, for whom 10% of the samples equal or exceed the HBM-GV_GenPop_. The RCRs calculated with the P95 results of Norwegian studies reported by Husøy et al. (2019) and Sakhi et al. (2018) [67,68] for adults are also lower than 1. However, the characterisation of risk is more uncertain because the results are reported as corrected by specific gravity. This makes it difficult to draw conclusions regarding risk because total BPS levels are very close to the HBM-GV. Similarly, for levels measured in children in the Sahki et al. (2018) study leading to a RCR that exceeds 1, the risk remains difficult to characterise, as these levels are adjusted for specific gravity and are close to the HBM-GV_GenPop_. 

##### For Workers

Based on the French study by Ndaw et al. (2018) on occupational biomonitoring for cashiers [69], RCRs were calculated considering the HBM-GV_worker_ value of 3.0 µg/L (Table 5). We can see that the RCRs are greater than 1 for cashiers and control workers not exposed to BPS in their occupational activities. The two median values (P50) for cashiers and controls do not exceed the HBM-GV_worker_ (3.0 µg/L). Therefore, according to the available results, although urinary total BPS levels are significantly higher in cashiers than in control workers, it appears that risk due to BPS exposure can be ruled out for at least 50% of the population in both groups, while for at least 5% of the same populations the risk exists. This also confirms that BPS exposure is ubiquitous in the general population and in workers.

## 4. Discussion

The expanding database on BPS provides convincing evidence of adverse effects on neurodevelopment or mammary glands following exposure during early life. The sensitivity to BPS is clearly dependent on the timing of exposure with specific periods of development being critical. Disruption of estrogenic signalling is likely central in the mediation of these effects although other modes of action may be involved [70,71,72]. The derived HBM-GV_GenPop_ is applicable to the whole general population and thereby is protective for all. Nevertheless, we are also aware of the current assessment of BPS at the EU level, which may be finalised during the coming months. Depending on the conclusion of this assessment, the derivation of the HBM-GVs for BPS could be updated accordingly. 

Consideration of the type and time of sampling is crucial, because the proposed HBM-GVs should allow for direct comparison with urinary data on bisphenols collected from the general population or occupational biomonitoring studies. It has to be noted that exposure is likely to be overestimated if spontaneous urine is used because spot urine sampling includes more sources of variability (e.g., within-person variability, within-day variability) than 24 h urine sampling, and that generic values in that case are generally used for the urinary output rate and bodyweight. 

Additionally, 24 h urine collections would be preferable as both the urinary concentration of total BPS or total BPA, and the urinary output rate (mL/day) are measured. The geometric mean (GM) (or median) can be directly used as an estimate for the average exposure of the population. The P95 might tend to overestimate high exposure as it includes not only the between-person variability, but also within-person and between-day variability [2]. However, collecting 24 h urine voids in large biomonitoring studies is not established and too challenging, mainly for reasons of logistics and cost. The collection of first morning urine is a non-random, single sampling that is not representative of daily variability, may introduce a bias and may result in an over- or underestimation of average exposure; however, the sparse evidence available from the literature does indicate comparability of the central tendency between first morning voids, spot urine samples and 24 h urine samples [2,73,74]. The total urinary concentration of BPS (or total BPA) in an individual spot urine sample cannot be used to arrive at a realistic estimate of daily exposure to bisphenols because of their non-persistent nature and short elimination half-life [2]. However, a set of spot urine samples can be used to obtain a reliable estimate of the average BPS exposure of a sufficiently large investigated population, provided that the sampling is at random in relation to meal ingestion and bladder-emptying times [2]. Vernet et al., who in 2018 characterised the within-day, between-day and between-week variability of phenol urinary biomarker concentrations (e.g., BPA) during pregnancy, came to the same conclusion that for biomonitoring purposes (and not aetiological studies) collecting spot samples was a good option if the population was large enough [74].

Regarding the assessment of risk due to exposure to BPA and BPS, the heterogeneity in terms of concentration units (i.e., µg/L, µg/g creatinine or µg/L corrected for urine specific gravity), statistical descriptors to document the distribution of exposure, as well as methodological sensitivity threshold limits of detection/limits of quantification and “quality assurance/quality control” provisions implemented to ensure consolidated consistency of the results generated, do not allow for direct and rapid comparability between different biomonitoring studies identified in the literature. The diversity of the subpopulations considered, the geographical sampling areas and the sampling years also contribute to the difficulty of comparing the reported data.

We find a clear difference between risk assessments for BPA and BPS exposure according to our current HBM-GVs. Indeed, according to current knowledge, while risk could be ruled out for BPA with very low RCRs for the general population, this is not the case for BPS where RCRs are high for a large portion of the sampled population, and it appears that protective measures need to be taken regarding BPS exposure in the general and occupational population. This is because the HBM-GVs for BPS were established on the basis of endocrine disruption effects in animals at very low doses, in contrast to the HBM-GV_GenPop_ for BPA. As mentioned in the introduction, the EFSA CEP Panel recently released a draft opinion for consultation regarding the re-evaluation of the t-TDI for BPA set in 2015 in light of the latest available data [19]. The new TDI would be established at 0.04 ng/kg bw/d of total BPA, which is 10^5^ times lower than the current t-TDI. As the HBM-GV is calculated under a steady-state, the new HBM-GV_GenPop_ for BPA based on the full new TDI using PBPK modelling would be 2.3 ng total BPA/L urine for adults and 1.4 ng total BPA/L urine for children assuming 100% oral intake in each case. Therefore, if this new TDI is confirmed after the consultation period, comparing the P95 for total BPA measured in the aligned studies to a HBM-GV derived from the new TDI proposed would result in RCRs far exceeding 1. Thus, the exposure to BPA would be of concern regarding this new value proposed, and protective measures would need to be taken. The possible concerns of citizens and stakeholders regarding their exposure over time to BPA doses would need to be considered. Such low levels of HBM-GVs may also be below the detection limits of most existing analytical methods, posing the challenge of further improving the sensitivity of detection. 

The HBM-GVs developed in the HBM4EU joint programme are associated with confidence levels based on expert judgments regarding the reliability of the data and the calculation method used to derive the HBM-GVs, also taking into account: the nature and quality of the epidemiological and/or toxicological data, uncertainties regarding the modes of action, the choice of starting point, as well as confidence in the PBPK modelling and the exposure scenario assumptions. The confidence levels for the HBM-GVs developed for measuring BPS for the general population and for workers were considered to be medium to low. These HBM-GVs thus need further scientific evaluation using a weight-of-evidence approach to be validated or to improve their confidence level. Several initiatives are underway and, depending on their findings, the risk assessment proposed here may change.

With respect to occupational populations, more studies need to be conducted to assess the risk of exposure to bisphenols for the European population for various occupational activities. This is true for BPA for which data are still limited. This is even more important for BPS and other BPA analogues in a context of BPA replacement in some products following the restriction of BPA in thermal papers, but also the evolution of societal concerns. Furthermore, there is a lack of knowledge at the international level regarding the actual exposure to BPA according to occupational activities, which triggers the need to perform biomonitoring studies for each activity. Calculating the difference in total BPA in urine from pre- and post-work samples may help to assess exposure resulting from workplace BPA ingestion at the individual level. This recommendation is also valid for BPS.

Overall, the available biomonitoring data indicate that the risk from occupational exposure should not be overlooked, although in the absence of a larger data set and a recommendable HBM-GV_worker_ for BPA it is difficult to draw a conclusion regarding risk to workers at this time.

## 5. Conclusions

In conclusion, this exercise exemplifies how biomonitoring data can be used in risk assessment associated to exposure to bisphenols, and also highlights a number of challenges and uncertainties related to this particular case study. The results point out the need to harmonise the process for collecting and expressing aggregated data, and thus to establish recommendations for conducting these biomonitoring studies, as well as the need to develop a network of laboratories with harmonised analytical capabilities to generate comparable exposure data. All these needs have been addressed in HBM4EU and should be further developed and improved in the future. There is also a need to bridge gaps in terms of knowledge of the toxicity of bisphenols A and S, particularly at low doses. Exposure assessment at the European level needs to be harmonised to better assess the risks. The EFSA Panel has recently reviewed new evidence of BPA toxicity, and if the effects of BPA at very low doses as suggested is confirmed in the near future, then most of the European population would be at risk. It is therefore necessary to continue to produce harmonised and accessible biomonitoring data both for the general population and according to different occupational activities. This is particularly relevant as regulatory action has been taken to reduce BPA, but an increase has been observed in the use and applications of BPS and other analogues.

All of this highlights the importance of developing and harmonising the use of biomonitoring to inform public policy, as is done in European research projects such as the HBM4EU. The new European Partnership for the assessment of risks from chemicals (PARC) is also moving in this direction and is even more broadly involved in the development of all aspects of chemical risk assessment. 

## Figures and Tables

**Figure 1 toxics-10-00228-f001:**
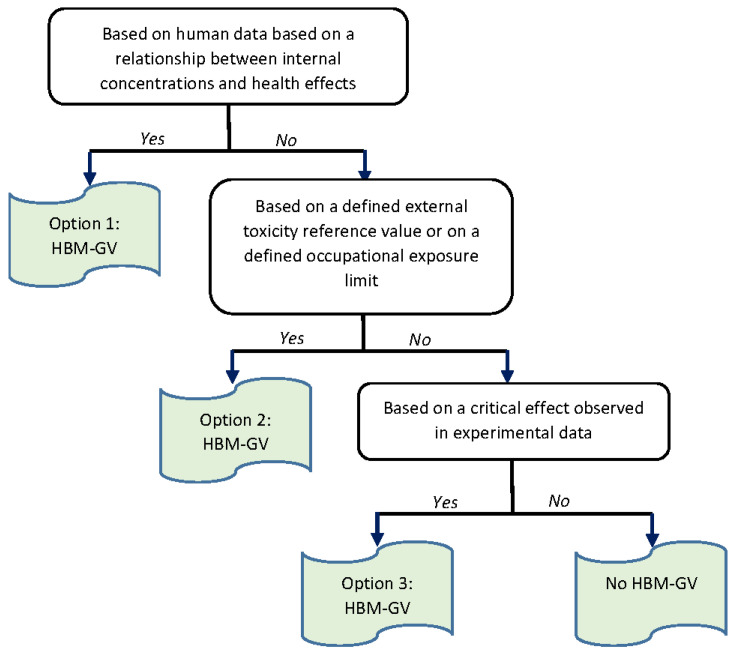
General methodology used to derive HBM-GVs as presented in Apel et al., 2020 [29].

**Figure 2 toxics-10-00228-f002:**
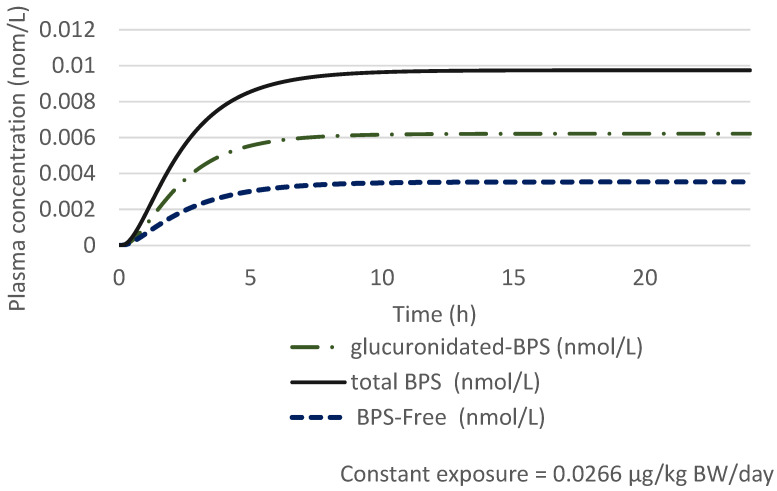
Estimated free BPS and glucuronidated BPS plasmatic concentrations (nmol/L) for a 70 kg adult after 24 h constant and continuous oral exposure to 0.0266 µg BPS/kg bw averaged over 24 h in the general population.

**Figure 3 toxics-10-00228-f003:**
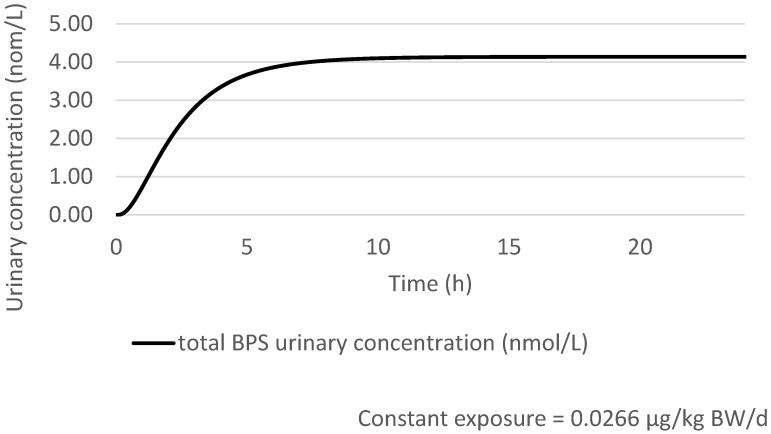
Estimated total urinary BPS concentration (nmol/L) for a 70 kg adult after 24 h constant and continuous oral exposure to 0.0266 µg of BPS/kg bw averaged over 24 h in the general population.

**Figure 4 toxics-10-00228-f004:**
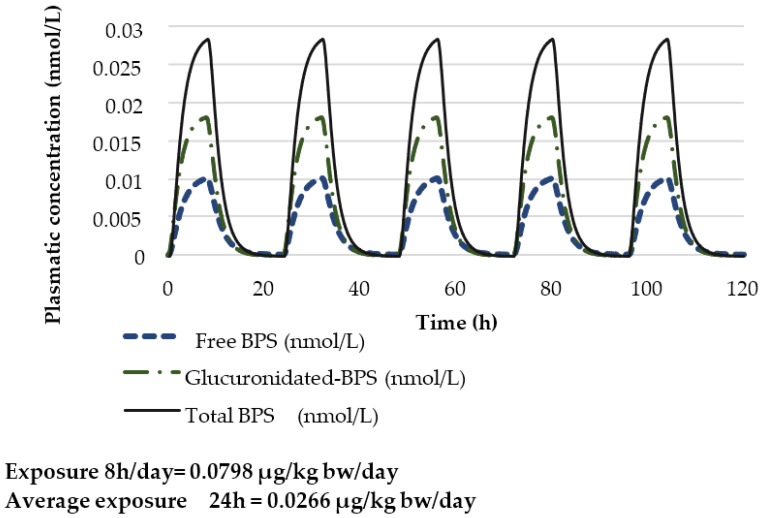
Estimated free BPS and glucuronidated BPS plasmatic concentrations (nmol/L) for an adult of 70 kg considering an occupational scenario lasting 5 days with daily exposure from 0 h to 8 h to 0.0798 µg/kg bw and from 8 h to 24 h to 0 µg/kg bw corresponding to an exposure for the entire day (0–24 h) of 0.0266 µg/kg bw/day.

**Figure 5 toxics-10-00228-f005:**
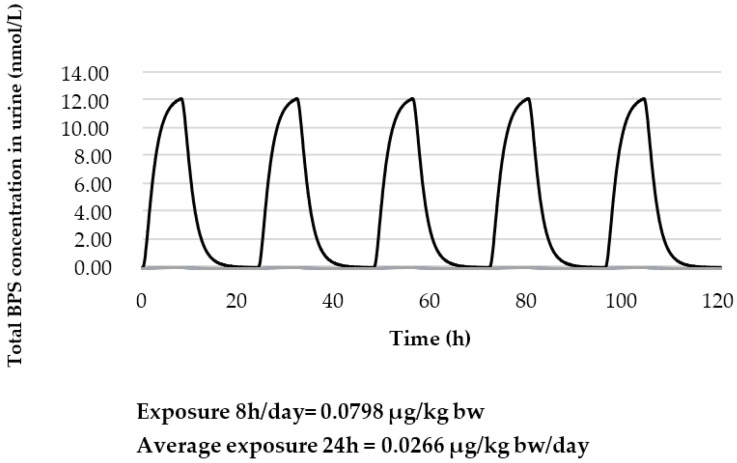
Estimated total urinary BPS concentrations (nmol/L) for an adult of 70 kg considering an occupational scenario lasting 5 days with daily exposure from 0 h to 8 h to 0.0798 µg/kg bw and from 8 h to 24 h to 0 µg/kg bw corresponding to an exposure for the entire day (0–24 h) of 0.0266 µg/kg bw/day.

**Table 1 toxics-10-00228-t001:** HBM-GVs calculated for the general population for total BPA in urine.

Key Study	Critical Effect	External Toxicity Reference Value	HBM-GV_GenPop_	Key Study
Tyl et al., 2008 (Two-generation toxicity study in mice)	Increase in relative mean kidney weight in adult males F0	t-TDI (EFSA, 2015) 4 µg/kg bw/day	230 µg/L	135 µg/L

**Table 3 toxics-10-00228-t003:** Key studies and selection of the POD for the derivation of the HBM-GVs.

Species, Exposure Duration	Critical Endpoint	POD	References
CD-1 Mice, oral exposure, GD 9 to PND 20	Increased number of terminal end buds in the mammary gland	LOAEL = 2 µg/kg bw/day	Kolla et al., 2018 [63]
CD-1 Mice, oral exposure, GD 9 to GD 16 or lactation day 20	Dose dependent increase in ductal area in the mammary gland.		Kolla et al., 2019 [64]
Rat, oral exposure, GD8/9 to PND20/21	Neurobehavioral toxicity		Catanese and Vandenberg, 2017 [65]

**Table 4 toxics-10-00228-t004:** Summary of HBM data and calculated RCRs for total urinary BPS for the general population.

Cohort	Reference	Country	Population	HBM-GV_GenPop_ (µg/L)	P25 (µg/L)	P50 (µg/L)	P75 (µg/L)	P90 (µg/L)	P95 (µg/L)	RCR
Esteban	Balicco et al., 2019 [49]	France	Adults (18–74 years old)	1.0	0.14	0.31	0.80	2.24	6.33	6.3
			Adult M (18–74 years old)	1.0	0.15	0.38	0.88	2.44	9.71	9.7
			Adult F (18–74 years old)	1.0	0.13	0.27	0.72	2.15	5.39	5.4
			Adults (18–29 years old)	1.0	0.16	0.44	0.99	3.38	7.13	7.1
			Adults (30–44 years old)	1.0	0.17	0.36	0.91	3.77	28.93	28.9
			Adults (45–59 years old)	1.0	0.14	0.29	0.7	1.97	3.33	3.3
			Adults (60–74 years old)	1.0	0.10	0.22	0.66	1.59	4.11	4.1
			Adults F (18–49 years old)	1.0	0.16	0.30	0.69	2.51	6.08	6.1
			Adults F (50 years and older)	1.0	0.11	0.23	0.78	1.98	3.49	3.5
Euromix	Husøy et al., 2019 * [67]	Norway	Adults (24–72 years old)	1.0	0.11	0.16	0.29	0.56	0.90	0.9
	Sakhi et al., 2018 * [68]	Norway	Mothers	1.0	<LOQ	<LOQ	0.26	0.44	0.59	0.6
			Children	1.0	<LOD	0.13	0.39	0.95	1.68	1.7
TH Pregnant women	Machtinger et al., 2018 [54]	Israel	Pregnant women	1.0		<LOD			0.40	0.4
SLO-CRP	Tkalec et al., 2021 [55]	Slovenia	Children (6–9 years old)	1.0		0.3		0.59	0.70	0.7
			Teenagers (11–15 years old)	1.0		<LOQ		1	1.80	1.8

* P95 reported as corrected by urine specific gravity.

**Table 5 toxics-10-00228-t005:** Summary of HBM data and calculated RCRs for total BPS for the occupational population.

Reference	Country	Population	HBM-GV_GenPop_ (µg/L)	P50 (µg/L)	P95 (µg/L)	RCR
Ndaw et al., 2018 [69]	France	Professional (cashiers)	3.0	2.53	19.9	6.6
		Professional (controls)	3.0	0.67	12.6	4.2

## Data Availability

Not applicable.

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
