# Peer review of "Human Biomonitoring Guidance Values (HBM-GVs) for Bisphenol S and Assessment of the Risk Due to the Exposure to Bisphenols A and S, in Europe"

_toxics, 2022, doi:10.3390/toxics10050228_

Round 1
Reviewer 1 Report
Overall, this paper provides an important contribution to the field of risk assessment for bisphenols. More information about BPA analogues such as BPS are critical to inform policy decisions.
The manuscript needs additional editing by someone with a very high proficiency in English or native English speaker. There are many word choice errors.
Structurally, I recommend some key revisions to improve comprehension to readers, and readability (easier to read). In many paragraphs, the most important fact/result/conclusion will come partway through the paragraph. A stronger, introductory, main sentence for each paragraph is needed. Key findings should be easier to find.
Please provide equations for calculations. For example, on line 433-434, authors mention the HBM-GV for workers is then rounded to 3 ug/L. I don't know how authors arrived at this conclusion from the previous figures presented.
Detailed editorial suggestions are proved in a separate file.

Author Response
We would like to thank the valuable comments of the reviewer.
Please see attachment for the responses.
Best regards,

Reviewer 2 Report
To my impression, this is a well-written, comprehensive and scientifically sound article on the use of HBM data in risk assessment of bisphenols A and S. From a regulatory point of view., the main weakness is the remarkable uncertainty with regard to the HBM-GV of BPA for people to whom exposure was mainly occupational. The HBM-GV for general population appears much more robust and reliable. However, the authors of this manuscript can in no way blamed for this uncertainty even though they might emphasize it a bit more. In the Discussion part, in contrast, uncertainties are adequately addressed. On balance, this manuscript is certainly worth to be published.
I have only minor (and more editorial) comments and remarks.
Introduction, lines 61 and 73: If you use the term "controlled", one might wonder if you mean that the risk was "acceptable" because exposure was well below TDI or DNEL. Might be the better description of the situation. "Insufficiently" controlled (line 73) might have to meanings: exceedance of DNEL or uncertainties with regard to exposure (estimates). Please clarify!
Lines 119 - 124: I wonder if classification of BPS as a reproductive toxicant has been already agreed on European level or if it is a proposal (by ...). Could you please clarify? The reader would think that this classification (legally binding or not) is based on the "fairly large body of data" as mentioned in the following sentence. It might be more logic to change the order of sentences in this paragraph, then.
Materials and methods, lines 147 - 151: Plese check the order of words in the first sentence ("can following" sounds odd), delete "according to HBM-GV derived" since this phrase is not necessary) and amend the apparently missing last words in the second sentence, following "measures".
2.2 General methodology: The approach published by Apel et al. (2020) is described here in brief. To my understanding, the different possibilities to derive HBM-GVs are rather "options" than "tiers" (see also line 186 where "option" is indeed used.) The latter suggests a stepwise approach. In reality, however, the decision for the most suitable option will depend on the database. Also, if I remember right, there was a third option in the cited article which might be also mentioned here, not only to be comprehensive and consistent but because it was used in case of BPS (as reported under 3.2.1.3).
Results, lines 176-178: It seems to me that this short paragraph was copy-pasted from somewhere else or is a remaining piece from a previous version. I suggest to delete it from the manuscript since it provides no useful information to the reader.
Line 190: Does the assumption of a 100% oral intake represent a worst case?
Line 197: Shouldn't it be eposure "of cashiers", instead of "for"?
Lines 198-208: In this whole paragraph, I miss an information regarding dermal absorption rate of BPA and the contribution of this rate to systemic or internal exposure. 100% skin exposure means only that dermal contact was the only exposure route but says nothing about the percentage entering the general circulation and resulting in a certain urinary excretion.
Lines 239/240: A slightly different wording is proposed that would better describe the approach in a more condensed sentence. Could be "RCRs were calculated by comparing total BPA levels in urine from the HBM studies to the corresponding HBM-GVs."
Lines 246-249: Please check the order of words. Also, the sentence might be kept shorter.
Table 2: What does "The 6 countries" really mean? Was it a meta-analysis of the six separate studies, putting all data together? What approach has been taken for this overall calculation? Please explain.
Line 255: "Occupational population" is (in contrast to "general population") at least an unusual term.
Line 259: What are "weak air levels"?
Lines 277-279: People can be exposed. "Certain professional activites such as..." cannot. Please correct.
3.2.1.3: The toxicological significance of the findings in mice (Table 3) is rather equivocal. The neurobehavioural findings in the study in rats should be reported in greater detail. Otherwise, one might consider the POD and resulting HBM-GV overly conservative and doubt the outcome of risk assessment.
Lines 471/472: Check the format please.
Conclusion, lines 620/621: I must confess not to understand what is meant with the half-sentence "low levels of hazard could be confirmed in the near future." Does it simply mean that the TDI might become very much lower and, accordingly, the results of RCR calculation will turn different?
Author Response
We would like to thank the reviewer for the valuable comments.
Please see attachment for the responses
Best regards,
